# Synergistic Effects of Functional CNTs and h-BN on Enhanced Thermal Conductivity of Epoxy/Cyanate Matrix Composites

**DOI:** 10.3390/nano8120997

**Published:** 2018-12-03

**Authors:** Mingzhen Xu, Yangxue Lei, Dengxun Ren, Sijing Chen, Lin Chen, Xiaobo Liu

**Affiliations:** Research Branch of Advanced Functional Materials, School of Materials and Energy, University of Electronic Science and Technology of China, Chengdu 610054, China; leiyangxue@163.com (Y.L.); rendenxun2008@126.com (D.R.); csj20189@126.com (S.C.); chenlin_uestc@163.com (L.C.)

**Keywords:** nanoparticles, copolymerization, functional CNTs, h-BN, cyanate ester

## Abstract

Epoxy/cyanate resin matrix composites (AG80/CE) with improved thermal conductivity and mechanical properties were obtained with synergetic enhancement with functional carbon nanotubes (*f*-CNTs) and hexagonal boron nitride (h-BN). AG80/CE performed as polymeric matrix and h-BN as conductivity filler which formed the main thermal conductivity channels. Small amounts of *f*-CNTs were introduced to repair defects in conductivity channels and networks. To confirm the synergetic enhancements, the thermal conductivity was investigated and analyzed with Agari’s model. Results indicated that with introduction of 0.5 wt% *f*-CNTs, the thermal conductivity coefficient (ƛ) increased to 0.745 W/mk, which is 1.38 times that of composites with just h-BN. Furthermore, the flexural strength and modulus of composites with 0.5 wt% *f*-CNTs were 85 MPa and 3.5 GPa. The glass transition temperature (*T_g_*) of composites with 0.4 wt% was 285 °C and the initial decomposition temperature (*T*_5%_) was 385 °C, indicating outstanding thermal stability. The obtained h-BN/*f*-CNTs reinforced AG80/CE composites present great potential for packaging continuous integration and miniaturization of microelectronic devices.

## 1. Introduction

The miniaturization and multi-functionality integration of electronic components have increased rapidly, and have demanded higher requirements on polymeric substrates because the rapidly increasing thermal accumulation of inner polymeric substrates can influence the performance and working reliability of substrate materials [1,2,3,4]. Thus, the designing and fabrication of polymeric composites with high thermal conductivities, satisfactory mechanical properties, and ideal thermal stabilities have become necessary. Moreover, it is also imperative to realize heat transfer by the substrate materials themselves, to prolong service life and avoid functional damage to components [5,6].

Epoxy resin and cyanate ester have been widely applied in the fields of electronics, automobile, sensor, aerospace, machinery, and chemical engineering, due to their excellent mechanical strength, friction resistance and electrical insulation, superior chemical resistance, good thermal and dimensional stability, and easy processing [7,8,9,10,11]. However, the relatively low thermal conductive coefficient (ƛ) of pristine epoxy resin and the cyanate ester matrix has limited their wider application [3,12]. 

As is well known, it is too difficult to fabricate a resin matrix with an intrinsic higher ƛ value than (>0.50 W/mK). Thus, thermally conductive inorganic fillers have been incorporated into polymeric composites to improve their thermal conductivity. As reported previously, addition of single thermally conductive fillers, including SiO_2_ [13], Al_2_O_3_ [14], BN [5,6,8], AlN [15], SiC [16], Si_3_N_4_ [17], CNTs [18], graphite nano-platelets [3,19], graphene oxide [1], graphene [11], etc. in epoxy resin matrices was investigated and results indicated that their thermal conductivities can be enhanced. Moreover, hybrid thermally conductive fillers, including Al_2_O_3_/AlN [20], AlN/BN, AlN/multi-wall carbon nanotubes (MWCNTs) [21], Cu/MWCNTs [22], SiO_2_/graphene oxide [23], BN/graphene oxide [24], graphite nanoplatelets/SiC [25], graphite nanoplatelets/CNTs [26], nanosilica/AgNWs, etc. were also introduced, and improved thermal conductivities of epoxy composites were obtained. Previous researches have achieved reasonable results. However, the researches were focused on category, shape, size, volume, and mass fraction, as well as surface functionalization of single and/or hybrid thermally conductive fillers on the thermal conductivities of epoxy composites, which limited their generalization in other high-performance resin matrices [2,9,11]. Furthermore, relatively high loading of thermally conductive fillers introduced into the resin matrix also results in adverse impacts on the processing behaviors and mechanical properties of the designed composites.

It is well known that the thermal conductivity of resin matrix composites with conductive fillers has been dominated by thermal conductivity channels and networks formed with fillers in the matrices [2,27]. Thus, the ability of fillers to form conductivity channels and networks as well as contact and overlap of fillers determined by the content of fillers, directly affects the thermal conductivity of the resulting composites [2,3,5,6,10]. In our previous work, epoxy/cyanate resin composites with various contents of h-BN were prepared and the results indicated that on increasing the content of h-BN, the thermal conductivity of the composites increased [28]. According to Agari’s model, with h-BN in the matrix it is hard to form effective conductivity channels [9]. Moreover, on continuous increase in the content of h-BN, the process behaviors of epoxy resin matrices worsen and it is difficult to obtain model composites. Additionally, the mechanical properties of the composites decrease significantly, which limits applications in the field of structural components. The decreases of mechanical properties has been mentioned in many reports [2,5,29].

In this work, it is proposed that in the case of a certain content of fillers in the resin matrix, improving the ability of fillers to form conductivity channels and networks can be expected to result in composites with satisfactory mechanical properties and improved thermal conductivity. Herein, functional CNTs (*f*-CNTs) and h-BN were introduced into the epoxy/cyanate resin matrix to obtain composites with improved combination properties. The effects of *f*-CNT content on thermal conductivity, mechanical properties, and thermal stability were investigated. Also, the synergetic enhancements of *f*-CNTs and h-BN on combination properties are also discussed.

## 2. Materials and Methods

### 2.1. Materials

CNTs (diameters: 10–30 nm, purity: >95%) were supplied by Chengdu Organic Chemistry Co. Ltd., Chinese Academy of Sciences, Chengdu, China. Carboxyl functionalized CNTs (*f*-CNTs) were prepared as previously reported [30]: raw CNTs were sonicated in a mixture of concentrated sulfuric and nitric acid (3:1 by volume) for 1 h at 60 °C. The solid was centrifuged and dried in a vacuum. The bisphenol A-based cyanate ester resin, 4,4′-(propane-2,2-diyl)bis(cyanatobenzene), was supplied by Wuqiao Resins Co. Ltd. (Yangzhou, China). Epoxy resin, AG80 (Equal Molar Weight (EMW) = 117–134 g/mol) was obtained from Shanghai Huaxi Resins Co. Ltd. (Shanghai, China). Hexagonal boron nitride (h-BN) powders were purchased from Macklin and used as received without further purification. Imidazole was used as initiator. Moreover, the thermal conductivity coefficients of the starting materials used in this work are presented in Table 1.

### 2.2. Preparation of the h-BN/f-CNTs Reinforced Matrix Blends

The required amount of epoxy resin and cyanate with a weight ratio of 4:6 was mixed and melted at 100 °C, with 0.5 wt% imidazole added as the latent initiator. Then, 40 wt% (weight of h-BN to the matrix) content of h-BN was added, with stirring for another 1 h. Various amounts of functional CNTs (*f*-CNTs) were added into the viscous solution with rapid stirring for another 2 h. The *f*-CNTs proportions were as follows: 0 wt%, 0.3 wt%, 0.4 wt%, and 0.5 wt% (weight of CNTs to the matrix), respectively. Then, the h-BN/*f*-CNT reinforced matrix blends were poured into molds and rapidly cooled to room temperature. The various blends were labeled as h-BN/*f*-CNTs/Matrix-0, h-BN/*f*-CNTs/Matrix-0.3, h-BN/*f*-CNTs/Matrix-0.4, h-BN/*f*-CNTs/Matrix-0.5, respectively, according to the various contents of *f*-CNTs.

### 2.3. Preparation of the h-BN/f-CNTs Reinforced Matrix Composites

First, a polytetrafluoroethylene mold with cavity dimensions 50 mm × 10 mm × 3 mm was preheated at 120 °C for 2 h. Then, the h-BN/*f*-CNTs/Matrix blending was melted at 100 °C for 20 min and the viscous melt was poured into the preheated polytetrafluoroethylene mold to carry out the standard procedure. The cured h-BN/*f*-CNT/Matrix composites were sanded to a thickness of 2 mm for dynamic mechanical measurements (DMA). Also, the cured h-BN/*f*-CNTs/Matrix composites were physically pulverized at ambient conditions for thermal gravimetric analysis (TGA).

### 2.4. Characterizations

Scanning electron microscope (SEM, JSM25900LV) was employed to observe the morphology of the functional CNTs and the fractured surfaces of the composites. X-ray photoelectron spectroscopic (XPS) measurements were carried out on an ESCA 2000 (VG Microtech, Uckfield, UK) using a monochromic Al Ka (hm = 1486.6 eV) X-ray source. Differential scanning calorimetric analysis (DSC) with a nitrogen flow rate of 50 mL/min and a heating rate of 10 °C/min was used to investigate the curing behavior of the blends. TGA was performed on a TA Instruments TGA Q50 with a heating rate of 20 °C/min (under nitrogen or air) and a purge flow rate of 40 mL/min. Thermal conductivity was measured with a Netzsch LFA 457 Laser Flash Apparatus. Mechanical properties were tested by flexural experiments with the three-point bending mode using the SANS CMT6104 series desktop electromechanical universal testing machine (CMT6104, Shenzhen, China) and a moving speed of crosshead displacement of 5 mm/min. Samples with dimension 50 mm × 10 mm × 1.2 mm were tested with the ratio of support span to thickness of 15:1. Results were obtained from the average value of three samples. Dynamic mechanical analysis (DMA) in a three-point-blending mode was performed on a QDMA-800 dynamic mechanical analyzer (TA Instruments, New Castle, DE, USA) to determine the glass temperature (*T_g_*). The storage modulus, tan delta, and glass transition temperature (*T_g_*) were studied with an amplitude of 15 µm and a frequency of 1 Hz, while the composites were all heated from 50 °C to 350 °C with a temperature ramp of 3 °C/min.

## 3. Results

### 3.1. Morphology and Structures of f-CNTs

Functional CNTs (*f*-CNTs) were prepared by acidizing and characterized mainly by SEM and XPS, shown in Figure 1 and Figure 2. In Figure 1a, the pristine CNTs are presented in large length-diameter ratios, which resulted in serious aggregation. It can be observed in Figure 1a that the aggregation is mainly ascribed to entanglement of the long CNTs. In comparison to that of pristine CNTs, *f*-CNTs show an obvious smaller length–diameter ratio and are homo-disperse without serious entanglement, shown in Figure 1b. Improved dispersion of *f*-CNTs was thus shown to be be conducive to the preparation of CNTs-reinforced matrix composites. 

To further characterize the structures, the obtained *f*-CNTs were characterized by XPS and TGA, as shown in Figure 2. It is obvious in Figure 2a, that the C1s spectrum of acidulated CNTs can be quantitatively differentiated into four different carbon species (O=C–O, C=O, C–O, and C– C/C=C). Moreover, the intensity of the C–O peak at 286.0 eV is marked, indicating successful preparation of *f*-CNTs [31]. Figure 2b presents TGA and DTG curves of pristine CNTs and *f*-CNTs. Pristine CNTs show excellent thermal stability and a slight decomposition appears above 600 °C. In comparison to that of pristine CNTs, *f*-CNTs show slight decomposition beginning at about 250 °C and an obvious weight loss appearing at about 310 °C, which can be assigned to the decomposition of oxygen functional groups, including hydroxyl, carboxyl groups, and others. DTG curves shown in Figure 2b, in which peaks appears at about 250 °C and 600 °C all correspond to changes of TGA curves. Moreover, on evaluating the decomposition processes of *f*-CNTs, the total weight loss shown in Figure 2b can also be assigned to the introduction of oxygen functional groups, that is, *f*-CNTs possess about 12 wt% oxygen functional groups. Thus, assisted by XPS tests and thermal stability measurements, the structures of *f*-CNTs were confirmed and the content of oxygen functional groups was also evaluated. Scheme 1 shows the preparation diagram of *f*-CNTs.

### 3.2. Curing Behaviors of h-BN/f-CNTs Reinforced Matrix Blends

According to our previous work, CE/AG80 blends filled with h-BN (40 wt%) were prepared and their curing behaviors characterized by DSC [28]. In this work, one of the aims was to reveal the influence of *f*-CNTs on the curing processes of the matrix resin. Thus, h-BN reinforced matrix resins with various contents of *f*-CNTs were investigated as shown in Figure 3. The shapes of DSC curves are usually used to reveal the curing processes, including the initial curing temperatures, peak curing temperatures, and post-curing temperatures. The detailed thermal properties of AG80/CE/h-BN with various contents of *f*-CNTs are collated in Table 2. In Figure 3, it is obvious that the main exothermic peak at about 250 °C can be assigned to copolymerization of CE and AG80 [32]. The approximate exothermic peaks indicate that the various contents of *f*-CNTs do not show any obvious influence on the curing behaviors of the resin matrix. Moreover, the fact that exothermic peaks appear at 320 °C, assigned to post-curing processes, also indicates that the main polymerizations of matrix resin are not affected by *f*-CNTs. However, compared with that of the resin matrix without *f*-CNTs, the initial curing temperatures of the resin matrix with *f*-CNTs shift to low temperature range, indicating that the *f*-CNTs may trigger copolymerization of CE and AG80. To summarize, the introduction of *f*-CNTs cannot significantly promote the main polymerization of resin matrix, but may trigger the reaction and initiate initial curing behavior in the low temperature range.

### 3.3. Thermal Conductivity of h-BN/f-CNT Reinforced Composites

The thermal conductivity (ƛ) values of h-BN/*f*-CNT reinforced resin matrix composites were investigated and the content of *f*-CNTs affecting the ƛ values of the resin matrix composites are shown in Figure 4. According to our previous work, the ƛ value of resin matrix with h-BN (40 wt%) is 0.54 W/mk [28]. Moreover, Agari’s model was used to evaluate the effect of the h-BN fillers in the resin matrix and results indicated that it was difficult to create thermal conductive channels efficiently under conditions, according to low C_2_. In Figure 4a, on introduction of *f*-CNTs, the ƛ values of composites increase significantly, composites with 0.5 wt% *f*-CNTs show a ƛ value of 0.745 W/mk. Besides, on increasing the content of f-CNTs, ƛ values increase correspondingly, increasing from 0.648 W/mk (with 0.3 wt% *f*-CNTs) to 0.745 W/mk (with 0.5 wt% *f*-CNTs). The microphotographs of h-BN/*f*-CNT reinforced composites are also presented in Figure 4c–f. Simulative thermal conductivity channels according to percolation theory are shown in the microphotographs. This indicates that with the introduction of bits of *f*-CNTs, resin matrix composites with improved thermal conductivity can be obtained. The reason can be attributed to various factors: first, nanoscale CNTs show outstanding thermal conductivity themselves; second, *f*-CNTs contribute to repair defects inside the h-BN filled resin matrix composites, due to the microscale of h-BN and its discoid shape. 

To confirm the effects of *f*-CNTs on forming thermal conductivity channels and networks of composites, Agari’s semi-empirical model is employed [9]. Agari’s model is based on the generalization of series and parallel conduction models in composites and correlates thermal conductivity with the ability of fillers to create particle conductive chains and is:
log*K_c_* = *φC*_2_log*K_f_* + (1 − *φ*)log*C*_1_*K_p_*
where *K_c_*, *K_f_*, and *K_p_* correspond to the thermal conductivity of composites, fillers, and polymer matrices, respectively. *φ* is the filler volume/weight fraction, *C*_1_ and *C*_2_ are obtained by fitting the experimental data. The logarithmic plot of thermal conductivity with respect to filler content is presented in Figure 4b, and the two parameters *C*_1_ and *C*_2_ were calculated to be 0.872 and 0.336. The value of *C*_1_ suggests that the introduction of *f*-CNTs affects the curing processes of the resin matrix, similar to that of h-BN and decreases the crosslinking degree of the matrices. The value of *C*_2_ indicates the ability to form conductivity channels and networks. In comparison with that of the resin matrix with pristine h-BN (*C*_2_ = 0.1593), the *C*_2_ (0.336) of composites with both h-BN and *f*-CNTs obviously increase. This visually shows that introduction of *f*-CNTs has improved the ability to form thermal conductivity channels and networks according to the percolation theory. Gu and his co-workers reported that combination of micrometer and nanometer BN fillers improved the thermal conductivity of resin matrix composites [2,5,6]. Thus, for CE/AG80/h-BN composites, the main heat transfer is dependent on conductivity channels forming with h-BN, which is defective due to the difficulty to form and stack-up discoid h-BN. After adding *f*-CNTs, conductivity channels and networks are easier to form and defective channels and networks will be repaired by *f*-CNTs, resulting in an improved thermal conductivity value [21,26].

Scheme 2 shows the possible route of forming improved conductivity channels and networks, with both h-BN and *f*-CNTs. As shown in Scheme 2a, primary conductivity channels are constructed with discoid h-BN. However, due to the microscale of h-BN, channels exist with obvious defects, that limit the improvement of the thermal conductivity. With the introduction of *f*-CNTs, the defects of the conductivity channels are repaired via interconnect effects of *f*-CNTs and h-BN shown in Scheme 2b. With contact of *f*-CNTs and h-BN, the channels and networks become perfect and thermal conductivity will increase correspondingly.

### 3.4. Fracture Surface of h-BN/f-CNT Reinforced Composites

Figure 5 shows the fracture surface of resin matrix composites with both h-BN and *f*-CNTs. Figure 5a, b show pristine h-BN and resin matrix. Obviously, discoid h-BN with uniform size of ~100 nm diameter, stacks, and aggregates are in view [28]. The pristine resin matrix shows a smooth fracture surface, indicating the typical brittle fracture. With addition of h-BN, the smooth fracture surface disappears as in Figure 5c. It has been reported that mutual contact and overlap of h-BN in the matrix exists and primary conductivity channels and networks are formed [2]. With addition of *f*-CNTs, the fracture surfaces of resin matrix with both h-BN and *f*-CNTs are different (Figure 5d–f), in which CNTs can be observed (labeled with red circles) and are dispersed uniformly. Combining the results of the thermal conductivity values of composites with *f*-CNTs shown in Figure 4a, the introduction of *f*-CNTs has contributed to forming perfect conductivity channels. Thus, the uniform dispersion of *f*-CNTs may just fix the defects resulting from random distribution of h-BN. Moreover, on increasing the content of *f*-CNTs, the CNTs in view increase as shown in Figure 5d–f.

### 3.5. Mechanical Properties of h-BN/f-CNTs Reinforced Composites

The mechanical properties of h-BN/*f*-CNTs composites determine their applications in the fields of functional-structural materials. Figure 6 shows the flexural strength and flexural modulus of the resin matrix with various contents of *f*-CNTs. It can be seen that the flexural strength of the pristine resin matrix composites shows an excellent value (72 MPa). With introduction of h-BN (40 wt%), the strength of the composites shows a slight decrease to 65 MPa. The strength decrease of the composites can be due to micro-scale h-BN in the resin matrix that may hinder polymerization and decrease the crosslinking degree. With addition of *f*-CNTs, the flexural strength of the composites increases correspondingly on increasing the content of *f*-CNTs. The obvious increase of flexural strength can be assigned to the reinforcement of nanoscale particles in the resin matrix. As is well known, the mechanical properties of resin matrix composites are mainly determined by the polymerization degree of the resin matrix and the dispersion of fillers in the resin matrix. Thus, in this work, the improvement of mechanical properties can be ascribed to the following factors: (1) the polymerization degree of composites may be improved with introduction of *f*-CNTs. As mentioned above, DSC results show that the initial polymerization temperatures of the resin matrix shift to low temperature ranges with the addition of *f*-CNTs, indicating the promotion effects of *f*-CNTs on the copolymerization of the matrix. (2) uniform dispersion of h-BN and *f*-CNTs may be also contributing to the perfect microstructures of composites, which will result in improved mechanical properties. By combining the fracture surface of the composites shown in Figure 5, it can be observed that h-BN and *f*-CNTs are uniformly dispersed in the matrix, which may synergistically improve the combination properties of the matrix composites. Figure 6 also shows the flexural modulus of composites with various contents of *f*-CNTs. Modulus values of composites increase correspondingly on increasing the content of *f*-CNTs. A similar variation trend of flexural strength and modulus also indicates that the introduction of *f*-CNTs significantly improves the mechanical properties of composites.

### 3.6. Thermal-Mechanical Properties of h-BN/f-CNTs Reinforced Composites

DMA measurements were applied to measure tan delta and the storage modulus. In Figure 7, the storage modulus and glass transition temperatures of h-BN/CE/AG80 composites with various *f*-CNTs contents are presented. Moreover, the values of the storage modulus at 50 °C are listed in Table 2. As reported, the dynamic properties represent the value of the energy dissipated in the strain process and the value of energy in the composite stored as elastic energy. On addition of fillers, geometrical characteristics, weight fraction, dispersion with resin matrix and load transfer from filler to matrix may influence the modulus to a great extent. In Figure 7a, the storage modulus of composites with 0.3 wt% and 0.4 wt% *f*-CNTs are higher than that of AG80/CE composites with just h-BN. The increase of storage modulus can be ascribed to the improvement of polymerization degree and the perfect microstructures of the composites. The composites with 0.5 wt% *f*-CNTs shows the lowest modulus, it may be attributed to aggregation of h-BN and *f*-CNTs, which introduces defects into the matrix. To summarize, all of the composites with both *f*-CNTs and h-BN show outstanding storage modulus and composites with moderate content of *f*-CNTs show higher modulus than that of composites with just h-BN, which also indicates synergetic enhancement of *f*-CNTs and h-BN on the thermal-mechanical properties of the composites.

Figure 7b presented plots of tan δ for composites with various contents of *f*-CNTs. It was reported that tan δ is usually used to reveal glass transition temperatures (*T_g_*) of resin matrix composites. Detailed data of the glass transition temperatures are summarized in Table 3. It is obvious that *T_g_s* show similar values (~260 °C) for composites with various contents of *f*-CNTs, indicating that the *T_g_s* of resin matrix composites have not been affected significantly by fillers. In comparison, composites with 0.4 wt% *f*-CNTs show the highest *T_g_* value (285.2 °C), indicating that addition of *f*-CNTs can improve the thermal property of composites to some extent.

Thermal stabilities of resin matrix composites with various contents of *f*-CNTs were characterized by TGA as shown in Figure 8 and the detailed data are collated in Table 3, in which the temperatures at weight loss of 5% (*T*_5%_), 10% (*T*_10%_), and char yield at 600 °C are displayed. According to Figure 8, it is obvious that composites with both *f*-CNTs and h-BN show higher decomposition temperatures than that of composites with just h-BN. Moreover, on increasing the content of *f*-CNTs, the decomposition of composites is almost constant. It can be explained by the fact that the addition of *f*-CNTs has improved the polymerization degree of the resin matrix but the content of *f*-CNTs shows no obvious effects on the copolymerization degrees. Additionally, the DTG curves of composites are also presented in Figure 8. The curve shapes are similar, indicating a similar decomposition mechanism. As reported, it can be concluded that introduction of *f*-CNTs will increase the copolymerization degree to some extent but not change the decomposition process of AG80/CE composites.

## 4. Conclusions

Synergetic enhancement of *f*-CNTs and h-BN on the combination properties of AG80/CE matrix composites was confirmed by investigation of thermal conductivity values, flexural strength and modulus, thermal-mechanical properties, and thermal stability. Results indicated that on introduction of *f*-CNTs, ƛ values increase by 38% from 0.54 W/mk to 0.745 W/mk for composites with 0.5 wt% *f*-CNTs. Agari’s model also shows that *f*-CNTs are conducive to forming conductivity channels and networks. Flexural strength (65–85 MPa) and modulus (2.7–3.5 GPa) also increase on introduction of *f*-CNTs, indicating enhancement of nanoparticles on mechanical properties. Combining storage modulus (4.2–5.1 GPa) and *T_g_s* (255–285 °C) properties, the enhancement of *f*-CNTs on thermal-mechanical properties of composites is significant, especially for composites with 0.4 wt% *f*-CNTs having been introduced. In summary, various investigations on combination properties of resin matrix composites with both h-BN and *f*-CNTs were implemented and synergetic enhancement of h-BN and *f*-CNTs was confirmed. Moreover, AG80/CE resin matrix composites with both h-BN and *f*-CNTs possess good thermal conductivity and thermal-mechanical properties, and can be used as candidates for applications for functional and structural materials. On considering both thermal conductivity and thermal mechanical properties, the introduction of 0.4 wt% *f*-CNTs may be a good method to fabricate satisfactory composites.

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
