# Peer review of "Synergistic Effects of Functional CNTs and h-BN on Enhanced Thermal Conductivity of Epoxy/Cyanate Matrix Composites"

_nanomaterials, 2018, doi:10.3390/nano8120997_

Round 1
Reviewer 1 Report
The paper presents an analytical and experimental work on the thermal conductivity of highly filled thermoset resin. Experimental investigations including microscopy, XPS, bending tests, DMA, DSC, TGA, Laser Flash are carried out to compare experimental results.
The obtained thermal conductivity of 0.5 W/mK by adding 40 wt% hBN is low, in the literature exit significant higher values of similar materials. This fact requires critical discussion.
The authors claim to figure out a synergistic effect by adding small portion of MWCNTs. To justify this claim, the paper lacks of substantial information.
1) Thermal properties of the constituent’s neat resin, h-BN, CNT are mandatory
2) Thermal properties of the neat resin with only CNTs
The statements of the curing behavior are difficult to follow, because the baseline of all curves in Fig 3 should be the same, otherwise the total mass is maybe different and cause different temperatures and curing. The scale of the heat flow is missing in Fig. 3. Often the 1st deviation of the curves are employed for detailed discussion.
The submission has low scientific standard e.g. citing not the original works and authors, respectively. Referring to Agrai’s model requires “Agari Y, Ueda A, Nagai S. Thermal conductivity of a polymer composite. J. Appl. Polym. Sci. 1993;49(9):1625–34.” Is the original work is not known to the authors?
The obtained results need to be discussed in context more critical, e.g.:
On page 9 line 272 the authors wrote “Composites with 272 0.5 wt% f-CNTs shows the lowest modulus, it may be attributed to aggregation of h-BN and f-CNTs, 273 which introduces defects into matrix. When this statement is correct several consequences follow: the strength and the thermal conductivity should be lowered as well.
According to which standard the bending tests were performed?
Which was the specimen size, crosshead speed?
The role of the CNT on the glass transition temperature (page 10 and table 2) and DMA results (storage modulus) needs more explanation. Why does the shape of the curves change? The influence of CNT should be visible above Tg.
The present discussion in the context of all obtained result are not logic (e.g. storage modulus, strength, thermal conductivity, Tg, …)
Author Response
Reviewers 1
The paper presents an analytical and experimental work on the thermal conductivity of highly filled thermoset resin. Experimental investigations including microscopy, XPS, bending tests, DMA, DSC, TGA, Laser Flash are carried out to compare experimental results.
1.The obtained thermal conductivity of 0.5 W/mK by adding 40 wt% h-BN is low, in the literature exit significant higher values of similar materials. This fact requires critical discussion.
Authors’ answer: Thanks for your suggestion. What you concerned are also what we have studied. In our previous work, considering of the production cost, macroscopic h-BN was used to manufacture composites with improved thermal conductivity. Due to its aggregation of h-BN itself, composites with 40 wt% h-BN couldn’t create effective conductivity channels/paths, resulting to an unsatisfactory thermal conductivity value of 0.5 W/mK. Thus, in this work, f-CNTs was introduced to improve the ability of macroscopic h-BN to create thermal conductivity channels, and then show improved thermal conductivity.
2. The authors claim to figure out a synergistic effect by adding small portion of MWCNTs. To justify this claim, the paper lacks of substantial information.
1) Thermal properties of the constituent’s neat resin, h-BN, CNT are mandatory
2) Thermal properties of the neat resin with only CNTs
Authors’ answer: We are grateful for your comments. In the revised manuscript, the substantial information was replenished in Materials part and copied as follows.
Table 1 Thermal conductivity coefficients of initial materials
Samples | Thermal conductivity coefficients (ƛ, W/mk) |
Neat matrix resin | 0.26 [28] |
h-BN | 300 |
CNT | 6600 |
f-CNT-0.5 wt%/matrix resin | 0.32 [experiment data] |
As reported in our previous work, neat matrix resin showed a thermal conductivity value of 0.26 W/mk, which is unsatisfactory for applications. Resin matrix with 0.5 wt% functional CNTs shows a slightly increase in thermal conductivity. With the substantial information, it’s logical to conclude that a synergistic effect can be obtained by adding small portion of f-CNTs. Thanks again for the reviewers’ comments to help us improving the manuscript in logical rigor.
3. The statements of the curing behavior are difficult to follow, because the baseline of all curves in Fig 3 should be the same, otherwise the total mass is maybe different and cause different temperatures and curing. The scale of the heat flow is missing in Fig. 3. Often the 1st deviation of the curves are employed for detailed discussion.
Authors’ answer: We are so sorry to make you confused on the curing behaviors. In Fig 3, DSC curves were used to present initial curing temperatures of resin matrix and effects of f-CNTs on curing behaviors. Thus, initial curing temperatures were collected in Table 2 and exothermic enthalpy was compared in qualitative. To show curves clearly, results were vertically shifted, which would not affect the results of comparison. When the exothermic enthalpy of DSC curves were compared in qualitative, scales of the heat flow would not be necessary. Thus, in Fig 3 the scales of heat flow are canceled. Additionally, the 1st derivation of the curves are discussed and presented in Table 2, which was defined as initial curing temperatures. In the revised manuscript, initial temperatures were used to reflect the effects of f-CNTs on the curing behaviors of resin matrix. Thanks again for your comments.
4. The submission has low scientific standard e.g. citing not the original works and authors, respectively. Referring to Agrai’s model requires “Agari Y, Ueda A, Nagai S. Thermal conductivity of a polymer composite. J. Appl. Polym. Sci. 1993; 49(9):1625–34.” Is the original work is not known to the authors?
Authors’ answer: We are so sorry to make you so angry on the citing of references. In this manuscript, the recommended reference “Y. Agari, T. Uno, Estimation on Thermal Conductivities of Filled Polymers, J Appl Polym Sci, 32 (1986) 5705-5712” was cited. The corresponding models or empirical formula in the recommended reference were presented and discussed in detail, presenting more reference value for our work of AG80/CE resin matrix with h-BN and f-CNTs filled. Thanks very much for your suggestion and your help to improve our manuscript.
5. The obtained results need to be discussed in context more critical, e.g.:
On page 9 line 272 the authors wrote “Composites with 272 0.5 wt% f-CNTs shows the lowest modulus, it may be attributed to aggregation of h-BN and f-CNTs, 273 which introduces defects into matrix. When this statement is correct several consequences follow: the strength and the thermal conductivity should be lowered as well.
According to which standard the bending tests were performed?
Which was the specimen size, crosshead speed?
Authors’ answer: Thanks for your comments. In this work, the bending tests were performed by flexural experiments with mode of three-point bending by the SANS CMT6104 series desktop electromechanical universal testing machine (CMT6104, Shenzhen, China) and moving speed of the crosshead displacement is 5 mm/min. Samples with dimension 50 mm×10 mm× 1.2 mm were test with the ratio of a support span to thickness of 15:1. Results were obtained by the average value of three samples. Due to the fact that results of DMA reflected the properties resulting from microscopic structures and aggregation structures of the nano-fillers reinforced resin matrix composites, dispersions and aggregations of nano-fillers would dominate the final results. However, mechanical properties of composites were assigned to the macroscopic properties, which would be affected by various facts including the microscopic structures of composites, the external stress on composites and so on. In sum, the microstructure changes can be investigated with results of DMA, while, mechanical tests just reflect the macroscopic properties of composites, which can be applied in structural applications.
6. The role of the CNT on the glass transition temperature (page 10 and table 2) and DMA results (storage modulus) needs more explanation. Why does the shape of the curves change? The influence of CNT should be visible above Tg.
Authors’ answer: We are grateful for your comments. The discussion and analysis on DMA results were replenished in detail in revised manuscript. According to the theory of nano-fillers reinforced resin matrix composites, storage modulus would increase with increasing the content of nano-fillers to some extent. Excessive content of nano-fillers would damage modulus due to their aggregation and heterogeneous dispersion. In this work, with increasing the content of f-CNTs, storage modulus of composites increased firstly and then decreased, which followed the above mentioned theory. In Fig 7, composites with 0.5 wt% f-CNTs showed a changed shape of curve, which can be assigned to the converted crosslinking structures of resin matrix composites. For resin matrix composites, their glass transition temperatures and thermal stability are dominated by the crosslinking structures and crosslinking degree. Combined the results of thermal stability, composites with 0.5 wt% f-CNTs showed the lowest decomposition temperature at 5% weight loss. Thus, the decrease of Tgs for composites with 0.5 wt% f-CNTs and the change of curve shape can be assigned to the decrease of crosslinking degree and altering of crosslinking structures.
7. The present discussion in the context of all obtained result are not logic (e.g. storage modulus, strength, thermal conductivity, Tg, …)
Authors’ answer: In the revised manuscript, all of the discussion and analysis were checked and revised carefully to ensure accuracy. For the less rigorous expression, we also amended carefully and all of the changes have been marked in Red. We think the revised manuscript has improved.

Reviewer 2 Report
In their submission to Nanomaterials entitled "Synergistic effects of functional CNTs and h-BN on enhanced thermal conductivity of epoxy/cyanate matrix composites", the authors describe the preparation of a series of epoxy/cyanate resin composite matrix of functional CNTs (f-CNTs) and hexagonal boron nitride (h-BN) with improved mechanical properties and thermal conductivity. The pape is well written and conclusions are well supported by experimental evidences as shown along the manuscript. The aforementioned materials have been characterized in detail. I would like to recommend this paper for its publication in Nanomaterilas after a few considerations:
a)Check the English style as the manuscript contains some typos and long sentences which hamper the Reading of the text.A revisión by a native speaker is recommended.
b) Check the reference sectiosn as references do not contain the journail with the corresponding abbreviation.
c) Have the authors performed any conductivity study on these materials?
d) Check the style for citing temperatura values as it should be "30 degree symbol followed by C", with a space between the number and the degree symbol. See for expample in Table 2.
Author Response
Reviewers 2
In their submission to Nanomaterials entitled "Synergistic effects of functional CNTs and h-BN on enhanced thermal conductivity of epoxy/cyanate matrix composites", the authors describe the preparation of a series of epoxy/cyanate resin composite matrix of functional CNTs (f-CNTs) and hexagonal boron nitride (h-BN) with improved mechanical properties and thermal conductivity. The paper is well written and conclusions are well supported by experimental evidences as shown along the manuscript. The aforementioned materials have been characterized in detail. I would like to recommend this paper for its publication in Nanomaterials after a few considerations:
a) Check the English style as the manuscript contains some typos and long sentences which hamper the Reading of the text. A revision by a native speaker is recommended.
Authors’ answer: Thanks for your suggestion. The manuscript has been carefully revised according to the reviewers’ comments and proof-read to minimize problems on grammar, format, etc.
b) Check the reference section as references do not contain the journal with the corresponding abbreviation.
Authors’ answer: Thanks for your reminding and the references were checked carefully according to the requirement of the Journal. Some revised references were copied as follows:
1. Song, N.; Yang, J.; Ding, P.; Tang, S.; Shi, L., Effect of polymer modifier chain length on thermal conductive property of polyamide 6/graphene nanocomposites. Compos Part A-Appl S 2015, 73, 232-241.
2. Gu, J. W.; Meng, X. D.; Tang, Y. S.; Li, Y.; Zhuang, Q.; Kong, J., Hexagonal boron nitride/polymethyl-vinyl siloxane rubber dielectric thermally conductive composites with ideal thermal stabilities. Compos Part A-Appl S 2017, 92, 27-32.
3. Gu, J.; Yang, X.; Lv, Z.; Li, N.; Liang, C.; Zhang, Q., Functionalized graphite nanoplatelets/epoxy resin nanocomposites with high thermal conductivity. Int J Heat Mass Tran 2016, 92, 15-22.
4. Xingyi, H.; Chunyi, Z.; Pingkai, J.; Dmitri, G.; Yoshio, B.; Toshikatsu, T., Polyhedral Oligosilsesquioxane‐Modified Boron Nitride Nanotube Based Epoxy Nanocomposites: An Ideal Dielectric Material with High Thermal Conductivity. Adv Funct Mater 2013, 23, (14), 1824-1831.
c) Have the authors performed any conductivity study on these materials?
Authors’ answer: Yes. Conductivity studies on these materials including dielectric constant/loss and electrical conductivity were performed. Results were presented in Fig 1 and Fig 2. With introduction of functional CNTs, h-BN/epoxy/CE composites showed expected dielectric properties. With increasing content of f-CNTs, dielectric loss decreased first and then increased, which can be assigned to the decrease of inner defects and the increase of conductive medium (f-CNTs). While, dielectric constant of composites increased persistently with increasing content of f-CNTs, which can be attributed to the improvement of electrical conductivity. Fig 2 showed the electrical conductivity of h-BN/epoxy/CE composites as functions of frequency. It was obvious that electrical conductivity increased with increasing frequency. In the range of test frequency, the designed h-BN/f-CNT reinforced composites were still insulation.
Fig 1 Dielectric properties of h-BN/f-CNT reinforced composites
Fig 2 Electric conductivity of h-BN/f-CNT reinforced composites
d) Check the style for citing temperature values as it should be "30 degree symbol followed by C", with a space between the number and the degree symbol. See for example in Table 2.
Authors’ answer: Thanks for your kind reminding. The style for citing temperature values were carefully checked and revised thoroughly.

Reviewer 3 Report
The manuscript deals with synthesis and characterisation of new polymer matrix composites with multiple types of fillers for improvement their thermal conductivity. Although the paper is prepared in a logical and concise manner, several corrections and discussion is addressed to the authors.
1. The authors mention “forming networks and channels”, which, in fact, is described by a percolation theory, which is well known and widely applied, especially in characterisation of such heterogeneous structures with jump-wise change of properties. According to this, it is recommended to extend literature review by introducing a percolation theory, adding the most important studies related to the topic of the manuscript, and link with the performed research study.
2. The structures of polymeric solutions presented in section 2.1 are well known and do not give any scientific input, I suggest to remove them from the manuscript.
3. Fig. 4a should be enlarged since the microphotographs there are not visible properly.
4. The presented results allow for evaluation of positive influence of f-CNTs to the composite, however, considering rationality of this approach, what is the minimal content of f-CNTs to obtain a material with good thermal conductivity, i.e. what amount of f-CNTs creates effective thermal pathways in the material? How it is related with other measured quantities? The detailed comments on these questions will improve readability of the manuscript.
Author Response
Reviewers 3
The manuscript deals with synthesis and characterization of new polymer matrix composites with multiple types of fillers for improvement their thermal conductivity. Although the paper is prepared in a logical and concise manner, several corrections and discussion is addressed to the authors.
1.The authors mention “forming networks and channels”, which, in fact, is described by a percolation theory, which is well known and widely applied, especially in characterization of such heterogeneous structures with jump-wise change of properties. According to this, it is recommended to extend literature review by introducing a percolation theory, adding the most important studies related to the topic of the manuscript, and link with the performed research study.
Authors’ answer: Thank you for your advice and your suggestion has been adopted. In the revised manuscript, literature reviews on the percolation theory were replenished in Introduction part and copied as follows.
It was well known that thermal conductivity of resin matrix composites with conductive fillers was dominated by thermal conductivity channels & networks formed with fillers in matrices[2, 27]. Thus, ability of fillers to form conductivity channels & network and contact & overlap of fillers determined by content of fillers, will directly affect thermal conductivity of resulted composites[2, 3, 5, 6, 10]. In our previous work, epoxy/cyanate resin composites with various contents of h-BN were prepared and results indicated that with increasing the content of h-BN, thermal conductivity of composites increased[28]. According to the Agari’s model, h-BN in matrix is hard to form effective conductivity channels[9]. Moreover, continuous increasing the content of h-BN, processes behaviors of epoxy resin matrices worsen and it was difficult to obtain model composites. Additionally, mechanical properties of composites decreased significantly, that would limit applications in fields of structural components. The decreases of mechanical properties has been reported in many reports[2, 5, 29].
2. The structures of polymeric solutions presented in section 2.1 are well known and do not give any scientific input, I suggest to remove them from the manuscript.
Authors’ answer: Thanks for your suggestion and your suggestion was adopted in the revised manuscript. The structures of AG80, CE and Imidazole were removed.
3. Fig. 4a should be enlarged since the microphotographs there are not visible properly.
Authors’ answer: Fig 4 was enlarged in the revised manuscript and copied as follows.
Fig 4 The thermal conductivity of h-BN/f-CNTs reinforced composites: (a) the content of f-CNTs affecting on ƛ value of resin matrix composites and (b) logarithmic plot of ƛ values with respect to the content of f-CNTs (Agari’s model).
4. The presented results allow for evaluation of positive influence of f-CNTs to the composite, however, considering rationality of this approach, what is the minimal content of f-CNTs to obtain a material with good thermal conductivity, i.e. what amount of f-CNTs creates effective thermal pathways in the material? How it is related with other measured quantities? The detailed comments on these questions will improve readability of the manuscript.
Authors’ answer: We are greatly appreciated you for your comments. What you concerned is what we have been studied. In this work, introduction of 0.5 wt% f-CNTs in h-BN/AG80/CE composites showed significantly improved thermal conductivity, indicating that effective thermal pathways have been created in the composites. Moreover, it can be seen in Fig 4 (a) that thermal conductivity of composites with 0.5 wt% f-CNTs increased sharply, which also indicated the effective thermal channels were beginning to create. However, considering the structural strength and modulus when it was applied as structural materials. In this work, flexural strength/modulus and thermal mechanical properties of the composites were performed and discussed. As is well known, flexural strength/modulus could reflect macroscopic mechanical properties, while thermal mechanical tests can be used to analyze the movement of molecular chains on the micro level. Considering of the fact that microstructures of polymer matrix composites would dominate the macroscopic properties, results of thermal mechanical properties was selected as the key reference object. In Fig 7 (a), it was obvious that composites with 0.5 wt% f-CNTs showed the lowest initial storage modulus and glass transition temperature, indicating that the introduction of 0.5 wt% f-CNTs has damaged the aggregation of molecular chains and then the movement of molecular chains. Composites with 0.4 wt% f-CNTs introduced showed outstanding storage modulus and glass transition temperature, indicating the positive influence of f-CNTs on thermal mechanical properties. In sum, considering of both thermal conductivity and thermal mechanical properties, introduction of 0.4 wt% f-CNTs may be good method to fabricate satisfactory composites. In the revised manuscript, detailed comments on these questions were replenished in Conclusion part and copied as follows.
Synergetic enhancement of f-CNTs and h-BN on combination properties of AG80/CE matrix composites was confirmed by investigation of thermal conductivity values, flexural strength and modulus, thermal-mechanical properties and thermal stability. Results indicated that with introduction of f-CNTs, ƛ values increase 38% from 0.54 W/mk to 0.745 W/mk for composites with 0.5 wt% f-CNTs. The Agari’s model also shows that f-CNTs are conducive to form conductivity channels & networks. Flexural strength (65~85 MPa) and modulus (2.7-3.5 GPa) are also increase with introduction of f-CNTs, indicating enhancement of nanoparticles on mechanical properties. Combining storage modulus (4.2-5.1 GPa) and Tgs (255-285 °C) properties, enhancement of f-CNTs on thermal-mechanical properties of composites is significantly, especially for composites with 0.4 wt% f-CNTs introduced. In summary, various investigations on combination properties of resin matrix composites with both h-BN and f-CNTs were implemented and synergetic enhancement of h-BN and f-CNTs were confirmed. Moreover, AG80/CE resin matrix composites with both h-BN and f-CNTs possess good thermal conductivity and thermal-mechanical properties, which can be used as the candidates for applications of functional and structural materials. Considering of both thermal conductivity and thermal mechanical properties, introduction of 0.4 wt% f-CNTs may be good method to fabricate satisfactory composites.

Round 2
Reviewer 1 Report
Almost all tasks were incorporated.
Author Response
Reviewers’ comments: Almost all tasks were incorporated.
Authors’ answers: Thanks for your comments and help on the improvements of our manuscript.
Reviewer 3 Report
The authors introduced corrections according the provided comments, however, some of them need further minor corrections.
1. The added paragraph in the Introduction does not address to the percolation itself. The percolation theory should be discussed here directly in the context of the performed examinations.
2. Still, the quality of Fig.4 is inappropriate. I suggest to make an additional figure of microphotographs and enlarge them to the dimensions of plots in Fig.4.
Author Response
The authors introduced corrections according the provided comments, however, some of them need further minor corrections.
1. Reviewers’ comments: The added paragraph in the Introduction does not address to the percolation itself. The percolation theory should be discussed here directly in the context of the performed examinations.
Authors’ answers: Thanks for your time and work. We are sorry for the misunderstanding in the previous manuscript. In the revised manuscript, percolation theory was discussed in Results and Discussion part “Thermal conductivity of h-BN/f-CNT reinforced composites”. The replenished percolation theory discussion was also copied as follows.
……Besides, with increasing the content of f-CNTs, ƛ values increase correspondingly, increasing from 0.648 W/mk (with 0.3 wt% f-CNTs) to 0.745 W/mk (with 0.5 wt% f-CNTs). The microphotographs of h-BN/f-CNT reinforced composites were also presented in Fig 4 (c, d, e and f). Simulative thermal conductivity channels according to the percolation theory were shown in microphotographs. It indicates with introduction of bits of f-CNTs, resin matrix composites with improved thermal conductivity can be obtained. The reason can be attributed various factors: first, nanoscale CNTs show outstanding thermal conductivity itself; second, f-CNTs are contributed to repair defects insides h-BN filled resin matrix composites, due to the microscale of h-BN and its discoid shape. ……
……It visually descripts that introduction of f-CNTs has improved forming ability of thermal conductivity channels & networks according to the percolation theory. Gu and his co-workers reported that cooperation of micrometer and nanometer BN fillers has improved the thermal conductivity of resin matrix composites [2, 5, 6]. Thus, for CE/AG80/h-BN composites, main heat transfer is depended on conductivity channels forming with h-BN, which is defective due to the difficulty to form and the stack-up of discoid h-BN. After adding f-CNTs, conductivity channels & networks will be easier to form and defective channels & networks will be repaired by f-CNTs, resulting in an improved thermal conductivity value[21, 26].……
2. Reviewers’ comments: Still, the quality of Fig.4 is inappropriate. I suggest to make an additional figure of microphotographs and enlarge them to the dimensions of plots in Fig.4.
Authors’ answers: Thanks for your comments. In the revised manuscript, microphotographs were replenished and enlarged in Fig 4 as follows.
Fig 4 The thermal conductivity of h-BN/f-CNTs reinforced composites: (a) the content of f-CNTs affecting on ƛ value of resin matrix composites, (b) logarithmic plot of ƛ values with respect to the content of f-CNTs (Agari’s model), (c) microphotographs of h-BN reinforced composites, (d) h-BN/f-CNTs reinforced composites with 0.3 wt% f-CNTs, (e) with 0.4 wt% f-CNTs and (f) with 0.5 wt% f-CNTs